# Neural representation of time across complementary reference frames

Yangwen Xu[1,2]*, Nicola Sartorato[1,3,4], Léo Dutriaux[1,5], Roberto Bottini[1]*

[1]Center for Mind/Brain Sciences, University of Trento, Trento, Italy; [2]Max Planck Institute for Human Cognitive and Brain Sciences, Leipzig, Germany; [3]Institute of Neurobiology, University of Tübingen, Tübingen, Germany; [4]Werner-Reichardt Centre for Integrative Neuroscience, Tübingen, Germany; [5]Laboratoire d'Études des Mécanismes Cognitifs, Université Lumière Lyon 2, Lyon, France

## eLife Assessment

This study presents a **valuable** finding on the neural representation of time from two distinct egocentric and allocentric reference frames. The evidence is **solid** and largely supports the hypothesis, with one caveat that the task differences could impact the observed effects. The work will be of interest to cognitive neuroscientists working on the perception and memory of time.

*For correspondence:
xuya@cbs.mpg.de (YX);
roberto.bottini@unitn.it (RB)

**Abstract** Humans conceptualize time in terms of space, allowing flexible time construals from various perspectives. We can travel internally through a timeline to remember the past and imagine the future (i.e., mental time travel) or watch from an external standpoint to have a panoramic view of history (i.e., mental time watching). However, the neural mechanisms that support these flexible temporal construals remain unclear. To investigate this, we asked participants to learn a fictional religious ritual of 15 events. During fMRI scanning, they were guided to consider the event series from either an internal or external perspective in different tasks. Behavioral results confirmed the success of our manipulation, showing the expected symbolic distance effect in the internal-perspective task and the reverse effect in the external-perspective task. We found that the activation level in the posterior parietal cortex correlated positively with sequential distance in the external-perspective task but negatively in the internal-perspective task. In contrast, the activation level in the anterior hippocampus positively correlated with sequential distance regardless of the observer's perspectives. These results suggest that the hippocampus stores the memory of the event sequences allocentrically in a perspective-agnostic manner. Conversely, the posterior parietal cortex retrieves event sequences egocentrically from the optimal perspective for the current task context. Such complementary allocentric and egocentric representations support both the stability of memory storage and the flexibility of time construals.

## Introduction

How the brain represents time remains mysterious. In prospective timing, a starting point is defined beforehand, and the neural activity tracks the elapsed time like a stopwatch. This neural stopwatch is manifested in various forms, including ramping activity, sequential activity, and neural trajectories (e.g., see reviews by *Buonomano and Laje, 2010*; *Wittmann, 2013*; *Eichenbaum, 2014*; *Tsao et al., 2022*). However, a stopwatch can only track duration in real time. How can we escape from the present, being able to remember the past or imagine the future?

One solution, which might be unique to humans, is to conceptualize time in terms of space (i.e., the spatial construal of time; e.g., *Clark, 1973*; *Traugott, 1978*; *Lakoff and Johnson, 1980*). Within

**eLife digest** Rather than being confined to a perpetual present, humans can adopt flexible perspectives on time. We can place ourselves within past and future events (i.e., mental time travel), or step back and survey them as a whole from a distance (i.e., mental time watching). This flexibility may come from how we frame time using spatial frameworks, specifically allocentric (world-centered) and egocentric (self-centered) reference frames.

In spatial processing, allocentric and egocentric representations rely on different regions in the brain. The medial temporal lobe, including the hippocampus, supports allocentric representations, whereas the parietal cortex supports egocentric ones. These two representations can be transformed into one another. Egocentric inputs can be integrated into a unified allocentric map in the medial temporal lobe (bottom-up processing), while allocentric representations can be used to reconstruct diverse egocentric perspectives in the parietal cortex (top-down processing).

The same mechanisms also apply to temporal processing. The hippocampus may encode a stable, perspective-independent representation of event sequences (allocentric time), while the parietal cortex may represent these sequences in a perspective- and task-dependent manner (egocentric time).

To test this hypothesis, Xu et al. asked participants to view the same sequence of events from either an internal (mental time travel) or an external (mental time watching) perspective while undergoing fMRI scanning. Activity in the posterior parietal cortex increased with greater temporal distance in the external perspective but decreased in the internal perspective. In contrast, hippocampal activity increased with temporal distance in both perspectives. These findings suggest that the hippocampus stores event sequences in a stable, perspective-independent manner, whereas the posterior parietal cortex retrieves them flexibly, depending on perspective.

Overall, these results show how the human brain supports flexible thinking about time while maintaining stable memories. They align with conceptual metaphor theory from cognitive linguistics, which proposes that abstract domains such as time are structured through more concrete domains like space, thereby enabling flexible access. This study will be relevant across disciplines, including philosophy, psycholinguistics, evolutionary biology, and cognitive neuroscience, particularly those concerned with the interplay of space, time, and memory.

this framework, time is usually first segmented into events—the basic temporal entities that observers conceive as having a beginning and an end (*Zacks and Tversky, 2001*). These temporal entities are then ordered in space, such that events occurring at different times can be maintained in working memory, allowing them to be flexibly accessed from different perspectives and easily referenced during communication (e.g., *Casasanto and Boroditsky, 2008*; *Núñez and Cooperrider, 2013*; *Bender and Beller, 2014*; *Abrahamse et al., 2014*; *Figure 1A*). The two core temporal components—duration and sequence—can be readily represented in such time construals.

Unlike prospective timing tracking the continuous passage of time, durations in time construals are event-based (*Sinha and Gärdenfors, 2014*): the interval boundaries are constituted by events, and the event durations reflect their span (*Figure 1A*). Accumulating evidence suggests that distinct cognitive systems underlie these two types of duration (e.g., *Block and Zakay, 1997*). The motor and attentional system—particularly the supplementary motor area (SMA)—has been associated with prospective timing (e.g., *Protopapa et al., 2019*; *Nani et al., 2019*; *De Kock et al., 2021*; *Robbe, 2023*), whereas the episodic memory system—particularly the hippocampus—is considered to support the representation of duration embedded within an event sequence (e.g., *Barnett et al., 2014*; *Thavabalasingam et al., 2018*; see also the comprehensive review by *Lee et al., 2020*).

As for event sequence, more than one century ago, the philosopher John McTaggart proposed the distinction between two different descriptions of time: the A-series and the B-series (*McTaggart, 1908*). The A-series assumes a deictic center of time—the observer's subjective now—as the reference, and orders events as being in the past or the future according to this deictic center. The B-series concerns the order of events in a sequence regardless of the observer's subjective now. The distinction between A- and B-series has been widely echoed in various accounts of the temporal frames of

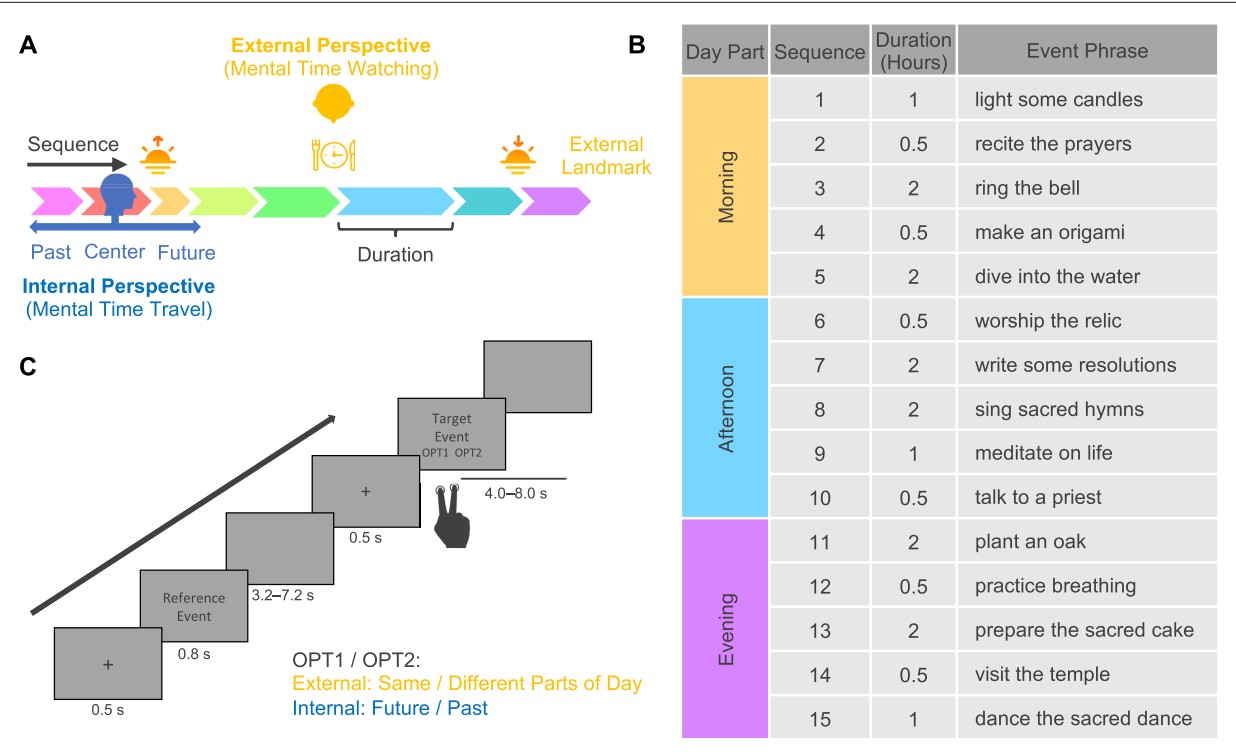

**Figure 1.** Spatial construal of time and experimental design. (**A**) The schematic diagram of spatial construal of time. It illustrates two core time concepts (sequence and duration) and two major perspectives on event series (mental time travel and watching). (**B**) Stimulus: a fictional religious ritual of 15 events following a specific sequence, enduring particular durations, and happening on predetermined parts of the day. To minimize potential confounds between the semantic content of the event phrases and the temporal structure of the events, we randomly assigned the phrases to the events, creating two versions for participants with even and odd ID numbers. Only one version is illustrated here. Both versions can be seen in *Figure 1—source data 1*. (**C**) Task paradigm. In the external-perspective task, participants judged whether the target events happened in the same part of the day as the reference event. In the internal-perspective task, participants imagined themselves doing the reference events and judged whether the target event happened in the past or will happen in the future.

The online version of this article includes the following source data for figure 1:

**Source data 1.** Event information and training materials.

reference under different terminologies in the cognitive linguistics (see reviews by *Bender and Beller, 2014*).

In general terms, these two construals of an event sequence can be understood through two complementary perspectives: an internal and an external one (*Núñez and Cooperrider, 2013*; *Tversky and Jamalian, 2021*; *Figure 1A*). The internal perspective on time is akin to the 'route' perspective in the spatial domain (*Siegel and White, 1975*). It aligns with the cognitive process called 'mental time travel' (*Tulving, 1984*; *Tulving, 2002*; *Suddendorf et al., 2009*). The traveler can project themselves into any event within the event sequence and redefine past and future according to their self-location—their subjective now. In this sense, the brain is no longer a stopwatch but a time machine, taking the traveler back and forth in time (*Buonomano, 2017*). The A-series is typically constructed from this internal viewpoint. By contrast, the external perspective on time is akin to the 'survey' perspective in the spatial domain (*Siegel and White, 1975*). It relates to the cognitive process called 'mental time watching' (*Stocker, 2012, Stocker, 2014*). The watcher, who is outside of the event sequence, can have a panoramic view of multiple events at different times and localize them relative to one another or to external temporal landmarks (e.g., sunrise and sunset in a day or historical events in the long term). In this sense, the brain is more like a dimensional ascension device, taking the watcher out of the one-dimensional timeline to an external viewpoint in higher-dimensional space. The B-series is generally constructed from this external viewpoint.

Recent studies have begun to investigate the neural representation of memorized event sequences (e.g., *Deuker et al., 2016*; *Thavabalasingam et al., 2018*; *Bellmund et al., 2019*; *Bellmund et al.,*

*2022*; see reviews by *Cohn-Sheehy and Ranganath, 2017*; *Bellmund et al., 2020*). However, the neural mechanisms that enable the brain to generate distinct construals of an event sequence remain largely unknown. Valuable insights may be drawn from research in the spatial domain, which posits the existence of stable allocentric representations that are independent of viewpoint, from which variable egocentric representations corresponding to different perspectives can be generated. According to an influential neurocomputational model (*Byrne et al., 2007*; *Bicanski and Burgess, 2018*; *Bicanski and Burgess, 2020*), allocentric and egocentric spatial representations are dissociable in the brain—they are respectively implemented in the medial temporal lobe (MTL)—including the hippocampus—and the parietal cortex. Various egocentric representations in the parietal cortex derived from different perspectives can be transformed and integrated into a unified allocentric representation and stored in the MTL (i.e., a bottom-up process). Conversely, the allocentric representation in the MTL can serve as a template for reconstructing diverse egocentric representations across different perspectives in the parietal cortex (i.e., a top-down process).

In line with the spatial construals of time hypothesis, several authors have recently suggested that such mutually engaged egocentric and allocentric reference frames (in the parietal cortex and the MTL, respectively) proposed in the spatial domain might also apply to the temporal one (e.g., *Gauthier and van Wassenhove, 2016a*; *Gauthier and van Wassenhove, 2016b*; *Gauthier et al., 2019*, *Gauthier et al., 2020*; *Bottini and Doeller, 2020*, *Bicanski, 2025*). If this hypothesis holds, it could explain how the brain flexibly generates diverse construals of the same event sequence. Specifically, the hippocampus may encode a consistent representation of an event sequence that is independent of whether an individual adopts an internal or external perspective, reflecting an allocentric representation of time. In contrast, parietal cortical representations are expected to vary flexibly with the adopted perspective that is shaped by task demands, reflecting an egocentric representation of time.

This functional magnetic resonance imaging (fMRI) study aimed to directly test this hypothesis by systematically investigating the neural mechanisms underlying the time construals of event sequence and duration. The event series was a fictional religious ritual of 15 events that participants learned the day before scanning (*Figure 1B*). Such a ritual had all the core temporal elements: the constituent events followed a specific sequence, endured particular durations, and happened on predetermined parts of the day (i.e., the external temporal landmarks). Participants learned these core temporal elements by reading the description and imagining going through the events one after another. The post-learning test suggests that all participants learned the temporal structure of the ritual before scanning (see 'Materials and methods' for details of the learning and testing procedure).

During the fMRI scanning, participants performed two tasks guiding them to consider the event series from internal and external perspectives (i.e., mental time travel vs. mental time watching; *Figure 1C*). In each trial, participants saw two sequential event phrases. The first was the reference event, and the second was the target event. In the external-perspective task, participants localized events relative to external temporal boundaries, judging whether the target event happened in the same or different part of the day as the reference event. In the internal-perspective task, participants were instructed to project themselves into the reference event and localize the target event relative to temporal point, judging whether the target event happened in the future or the past of the reference event (see 'Materials and methods' for details of the scanning procedure).

## Results

### Time was processed differently from internal and external perspectives

Participants had significantly greater accuracy in the external-perspective task than the internal-perspective task (external-perspective task: M=93.5%, SD = 4.7%; internal-perspective task: M=89.5%, SD = 8.1%; paired t(31) = 3.33, p=0.002). The reaction time (RT) of the corrected trials in the external-perspective task was also significantly shorter than the internal-perspective task (external-perspective task: M=1475 ms, SD = 529 ms; internal-perspective task: M=1578 ms, SD = 587 ms; fixed effect of *Task Type* in a random-intercept-and-slope linear mixed model (LMM) with *Participant* as the random-effect grouping factor: F(1, 31)=27.44, p<0.001).

To further explore the factors affecting the RT of the correct trials, we built a random-intercept LMM with *Participant* as the random effects grouping factor. Fixed effects variables included *Sequential Distance* (i.e., number of events between the reference and the target events), *Duration* (i.e.,

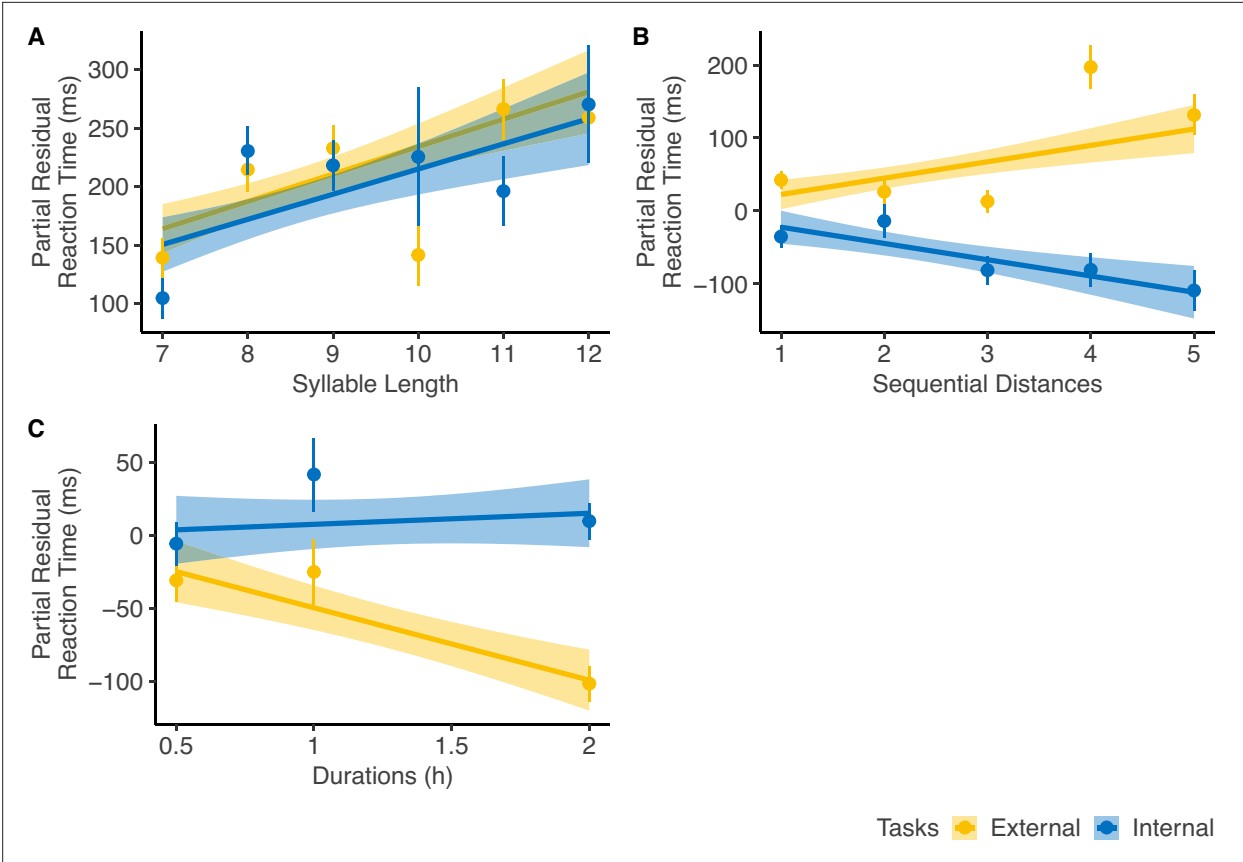

**Figure 2.** Reaction time (RT) analysis. For diagnostic purposes, we plotted the partial residuals of each significant predictor significantly influencing the RT of the corrected trials (n=32). The partial residual includes the effect of each variable, its interaction with *Task Type*, and the residuals from the full linear regression model. (**A**) RT increased with *Syllable Length*, showing a similar trend across both tasks. (**B**) *Sequential Distance* affected RT in opposite directions depending on the perspectives. (**C**) *Event Duration* influenced RT only in the external-perspective task, with no effect in the internal-perspective task. The error bar indicates the standard error relative to the mean, and the shaded band around the linear regression line indicates 95% confidence interval.

The online version of this article includes the following source data and figure supplement(s) for figure 2:

**Source data 1.** Behavioral data of the fMRI tasks.

**Figure supplement 1.** Reaction time (RT) analysis including parts of the day.

duration of the target events), *Syllable Length* (i.e., number of syllables of the phrase of the target events), *Task Type* (i.e., external- vs. internal-perspective tasks), and the interaction between *Task Type* and all the other variables (*Figure 2*; *Table 1*). As a sanity check, we found a significant main effect of *Syllable Length* (F(1, 6918)=35.59, p<0.001) since it was expected that participants spent longer time reading longer phrases. Intriguingly, we also found significant interaction effects between *Task Type* and *Sequential Distance* (F(1, 6918)=28.22, p<0.001) and *Task Type* and *Duration* (F(1, 6918)=12.81, p<0.001).

*Sequential Distance* was correlated positively with RT in the external-perspective task (z=3.80, p<0.001) but negatively in the internal-perspective task (z=−3.71, p<0.001). The negative correlation between RT and egocentric distance has been consistently observed in previous studies in which participants engaged in temporal self-projection: participants make more effort to differentiate past and future for the events close to their own temporal position (e.g., *Arzy et al., 2008*; *Arzy et al., 2009a*; *Arzy et al., 2009b*; *Gauthier and van Wassenhove, 2016a*; *Gauthier and van Wassenhove, 2016b*; *Gauthier et al., 2019*). This pattern can broadly be attributed to the symbolic distance (SD) effect (*Moyer and Landauer, 1967*; *Moyer and Bayer, 1976*; *Shepard and Judd, 1976*), which refers to the fact that "time needed to compare two symbols varies inversely with the distance between their referents on the judged dimension" (*Moyer and Bayer, 1976*, p. 229). The positive correlation

**Table 1.** The reaction time of the corrected trials indicates that time is differently processed under internal and external perspectives. Linear mixed model: RT ~ 1 + Task Type * (Sequential Distance + Duration + Syllable Length) + (1 | Participant).

| Fixed effects* | df | F | p |
|---|---|---|---|
| Sequential Distance | 1, 6918 | 0.0003 | 0.987 |
| Duration | 1, 6918 | 6.914 | 0.009 |
| Syllable Length | 1, 6918 | 35.585 | <0.001 |
| Task Type | 1, 6918 | 4.357 | 0.037 |
| Task Type × Sequential Distance | 1, 6918 | 28.224 | <0.001 |
| Task Type × Duration | 1, 6918 | 12.809 | <0.001 |
| Task Type × Syllable Length | 1, 6918 | 0.065 | 0.799 |

*The significant effects were highlighted. We did not highlight the significant main effects if the corresponding interaction effects were also significant.

between RT and sequential distance from an external perspective was predicted by *Gauthier and van Wassenhove, 2016a*. This prediction was inspired by classic studies on mental scanning using visual imagery (e.g., *Shepard and Judd, 1976*; *Kosslyn et al., 1978*): From an external perspective, people might compare two referents by mentally traversing the intermediate states between them, resulting in longer times to scan longer sequential distances.

As for *Duration*, it had a significantly negative trend with RT in the external-perspective task (z=–4.44, p<0.001) but not in the internal-perspective task (z=0.66, p=0.51). A possible explanation is that events with longer durations were more salient and thus easier to be compared with external landmarks.

To ensure that the behavior outputs in neither internal- nor external-perspective tasks confounded the effects above, we built another LMM incorporating four additional variables: Same/Different parts of the day, Future/Past, and their respective interactions with *Task Type* (*Figure 2—figure supplement 1*; *Supplementary file 1*). The interaction effects between *Task Type* and both *Sequential Distance* and *Duration* remained significant. Furthermore, we found an additional interaction effect between *Task Type* and *Same/Different* parts of the day (F(1, 6914)=70.49, p<0.001). This effect was in line with the effect of *Sequential Distance*, with two events in the same and different parts of the day corresponding to the short and long sequential distances, respectively. This effect paralleled that of Sequential Distance, as events occurring within the same or different parts of the day corresponded to shorter and longer sequential distances, respectively. This pattern can be interpreted as a categorical effect: sequential distances within the same part of the day were perceived as shorter (i.e., a chunking effect), whereas distances spanning different parts of the day were perceived as longer (i.e., a boundary effect). Similar boundary- or chunking-related effects on event cognition have been reported in previous studies (e.g., *Ezzyat and Davachi, 2011*; *DuBrow and Davachi, 2013*; *Radvansky and Zacks, 2017*).

The behavioral results suggest that our attempt to induce different perspectives on the event series was successful. The two tasks induced RT as distinct functions of sequential distance, in line with the predicted SD effect for the internal perspective and reverse-SD effect for the external perspective (*Gauthier and van Wassenhove, 2016a*).

## Internal- compared to external-perspective task activated different brain networks

We first directly contrasted the activity level between external- and internal-perspective tasks in the time window of the target events (*Figure 3A*; *Table 2*; see *Figure 3—figure supplement 1A* for a surface view; voxel level p<0.001, cluster-level Family-Wise Error [FWE] corrected p<0.05). Compared with the external-perspective task, the internal-perspective task specifically activated the regions in the default network (DN) in the right hemisphere. They were the precuneus (PreC), the retrosplenial cortex (RSC), the superior frontal gyrus (SFG), and the angular gyrus (AG). This finding aligned with evidence indicating that the DN plays a crucial role in self-projection (see the review by *Buckner and Carroll, 2007*). In *Figure 3—figure supplement 1*, we also compared the significant clusters

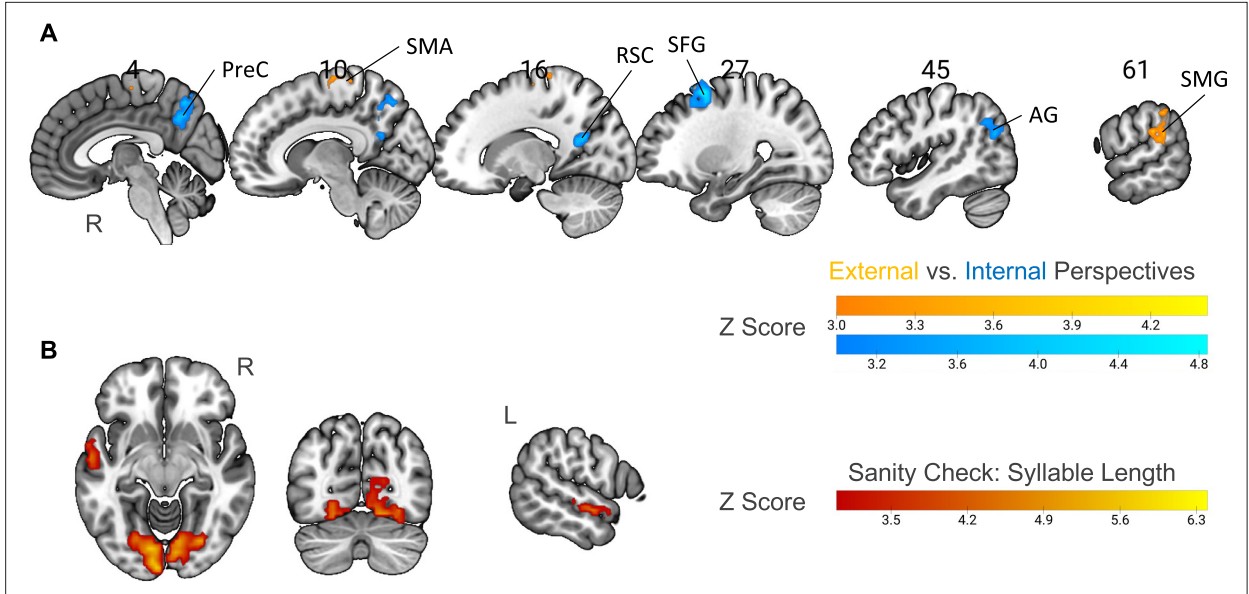

**Figure 3.** Neural correlates of specific perspectives and syllable length. (**A**) Univariate contrast between external-perspective and internal-perspective tasks (n=32; voxel-level p<0.001, cluster-level FWE-corrected p<0.05). All the significant areas were in the right hemisphere. PreC: precuneus; RSC: the retrosplenial cortex; SFG: the superior frontal gyrus; AG: angular gyrus; SMA: supplementary motor area; SMG: supramarginal gyrus. (**B**) Parametric modulation of syllable length as a sanity check (n=32; voxel-level p<0.001, cluster-level FWE-corrected p<0.05). The activation level in the anterior part of the left superior temporal gyrus and the visual cortex positively correlated with syllable length. R=right hemisphere; L=left hemisphere.

The online version of this article includes the following source data and figure supplement(s) for figure 3:

**Source data 1.** Thresholded Z-score maps of the univariate contrast between external- and internal-perspective tasks, and the parametric modulation of syllable length.

**Figure supplement 1.** Cortical surface visualization of neural correlates of distinct perspectives.

**Table 2.** Univariate contrast between internal- and external-perspective tasks (p<0.001, cluster-level FWE-corrected p<0.05 across the whole cortex).

| Anatomical label | Center MNI coordinate* | | | Cluster size |
| --- | --- | --- | --- | --- |
| | X | Y | Z | Number of voxels[†] |
| Internal > external-perspective task | | | | |
| Supplementary motor area (R) | 10 | −22 | 69 | 34 |
| Supramarginal gyrus (R) | 61 | −36 | 26 | 98 |
| Internal > external-perspective task | | | | |
| Precuneus (R) | 4 | −61 | 45 | 79 |
| Retrosplenial cortex (R) | 16 | −56 | 19 | 45 |
| Superior frontal gyrus (R) | 27 | 12 | 56 | 95 |
| Angular gyrus (R) | 45 | −66 | 30 | 78 |

*The average Montreal Neurological Institute coordinates of all the significant voxels of each cluster. The precuneus and the retrosplenial cortex were connected as one cluster under the threshold p<0.001 (z>3.09). In this case, we increased the threshold to the point when the precuneus and the retrosplenial cortex were separate (z > 3.3) and calculated the average coordinates of each cluster.

[†]The voxel size is 3 × 3 × 3 mm³.

in the internal-perspective task with the two subnetworks of the DN: DN-A and DN-B (e.g., *Braga and Buckner, 2017*; *Reznik et al., 2023*). The internal-perspective mostly engages DN-A rather than DN-B. This observation is consistent with existing evidence suggesting that DN-A is more closely associated with episodic memory, whereas DN-B is primarily involved in social processing (e.g., *Lin et al., 2018*; *DiNicola et al., 2020*).

Compared with the internal-perspective task, the external-perspective task specifically activated the SMA and the supramarginal gyrus (SMG) in the right hemisphere (*Figure 3A*; *Table 2*; voxel level p<0.001, cluster-level FWE-corrected p<0.05). This finding is open to interpretation. The area found in the right SMG was centered at the junction region between the posterior part of the SMG and the posterior part of the superior temporal gyrus. Previous evidence has shown that this temporoparietal junction area relates to the out-of-body experience (e.g., see reviews by *Blanke et al., 2004*; *Blanke and Arzy, 2005*) or plays a role in overcoming egocentric emotion bias toward others (e.g., *Silani et al., 2013*). Thus, the right SMG region found here may be important for the mental construction of an external perspective.

## The right posterior parietal cortex implemented opposite representations of sequential distance across external and internal perspectives

Next, we used the parametric modulation method to detect neural correlates of the temporal information (i.e., *Sequential Distance* and *Duration*) across external- and internal-perspective tasks. We built a single general linear model (GLM) in which the target events were simultaneously modulated by their duration and their sequential distances to the reference events. The target events in external- and internal-perspective tasks were treated separately. To detect whether the temporal information was represented differently from different perspectives, we first examined the interaction effect between each temporal information and the task type. If the interaction effect was not significant, we also examined the main effect combining both tasks.

As a sanity check, we investigated whether the employed parametric modulation method could successfully reveal the effect of the word length in the visual cortex, given that our stimuli were visually presented. To do so, we used *Syllable Length* as the parameter to modulate the condition of both the reference and the target events in the above GLM. We found not only the visual cortex but also the left superior temporal gyrus in the language network, of which the activity level positively correlated with the number of syllables (*Figure 3B*; voxel level p<0.001, cluster-level FWE-corrected p<0.05). This result confirmed our prediction and validated our methodology.

To detect the regions in which the activation level was modulated by *Sequential Distance* (i.e., the number of events between the reference and the target events), we first searched regions with a significant interaction effect between *Task Type* (i.e., external- vs. internal-perspective tasks) and *Sequential Distance* (*Figure 4A–C*). The only significant region we could find across the whole cortex was localized in the border area between the angular gyrus and the superior division of the lateral occipital cortex in the right hemisphere (i.e., the boundary area between Brodmann area 7, 19, and 39; *Figure 4A*; voxel level p<0.001, cluster-level FWE-corrected p<0.05). More specifically, this region is mostly at the lateral wall of the posterior intraparietal sulcus (hIP5: 56.2%, hIP6: 9.5%, hIP4: 5.9%; *Richter et al., 2019*) and the posterior part of the angular gyrus (PGp: 21.4%; *Caspers et al., 2006*; *Caspers et al., 2008*) (assignment based on maximum probability map; *Eickhoff et al., 2005*). The MNI coordinate of the center voxel is 38, –69, 35. For convenience, we will refer to this region as the posterior parietal cortex (PPC) in the following text. *Figure 4B* further shows the beta estimates of *Sequential Distance* in the parametric modulation analysis in the right PPC. The activation level in this region correlated with sequential distance positively in the external-perspective task (t(31) = 2.97, p=0.006) but negatively in the internal-perspective task (t(31) = –4.19, p<0.001).

Here, changes in activity levels within the PPC were found to align with RT. Whether to control for RT's influence on fMRI activation represents a well-known paradox. On the one hand, RT reflects underlying cognitive processes and therefore should not be fully controlled for. On the other hand, RT can independently influence neural activity, as several brain networks vary with RT irrespective of the specific cognitive process involved—a domain-general effect. For instance, regions within the multiple-demand network are often positively correlated with RT and task difficulty across diverse cognitive domains (e.g., *Fedorenko et al., 2013*; *Mumford et al., 2024*). To evaluate the second possibility, we

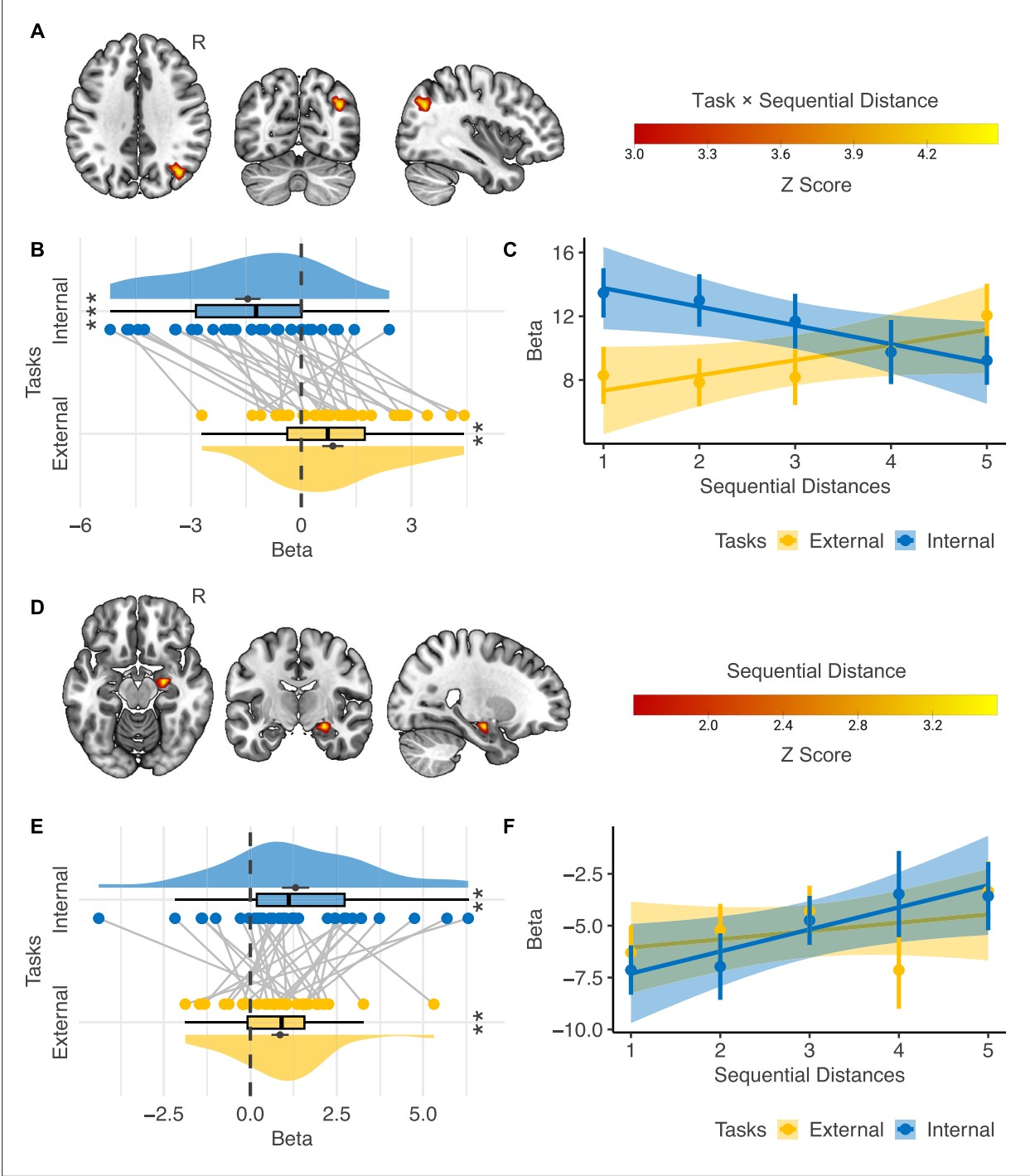

**Figure 4.** Neural correlates of event sequence. (**A–C**) Interaction effect between *Task Type* (i.e., external- vs. internal-perspective tasks) and *Sequential Distance* (n=32) .(**A**) The only cortical region showing a significant interaction effect was localized in the right posterior parietal cortex (voxel-level p<0.001, cluster-level FWE-corrected p<0.05). (**B**) Regions of interest analysis shows that the activation level in the right posterior parietal cortex correlated with sequential distance positively in the external-perspective task and negatively in the internal-perspective task. (**C**) A further illustration of the relations between the activation level in the right posterior parietal cortex and sequential distance in the two tasks. The error bar indicates the standard error relative to the mean, and the shaded band around the linear regression line indicates 95% confidence interval. (**D–F**) Main effect of *Sequential Distance* (n=32). (**D**) The right hippocampal head shows a significant main effect of *Sequential Distance* within the mask of the bilateral hippocampus (voxel-level p<0.001, cluster-level FWE-corrected p<0.05; voxel-level p<0.05 for illustration purposes). (**E**) Regions of interest analysis shows that the correlation between the activation level in the right hippocampal head and the sequential distance was independent of perspectives. (**F**) A further illustration of the relations between the activation level in the right hippocampal head and sequential distance in the two tasks. The error

*Figure 4 continued on next page*

*Figure 4 continued*

bar indicates the standard error relative to the mean, and the shaded band around the linear regression line indicates 95% confidence interval. R=right hemisphere. **p<0.01; ***p<0.001.

The online version of this article includes the following source data for figure 4:

**Source data 1.** Thresholded Z-score maps of the neural correlates of event sequence and the extracted beta values for ROI analysis.

conducted an additional control analysis by including trial-by-trial RT as a parametric modulator in the first-level model (see 'Materials and methods'). Notably, the same PPC region remained the only area in the entire brain showing a significant interaction between *Task Type* and *Sequential Distance* (voxel-level p<0.001, cluster-level FWE-corrected p<0.05). This finding indicates that PPC activity cannot be fully attributed to RT. Furthermore, we do not interpret the effect as reflecting a domain-general RT influence, as regions within the multiple-demand system—typically sensitive to RT and task difficulty—did not exhibit significant activation in our data.

To further assess whether the observed PPC reactivation can be attributed to boundary or chunking effects introduced by *the Parts of the Day,* as well as other behavioral outputs, we performed an additional control analysis. Using a more complex first-level model, we included two extra regressors modulating the target events in both internal- and external-perspective tasks, alongside *Sequential Distance* and *Duration*: (1) Same/Different parts of the day (coded as 1/–1) and (2) Future/Past (coded as 1/–1). Even with these additional controls, the same PPC region remained the strongest area across the entire brain, showing an interaction effect between *Task Type* and *Sequential Distance*, although the cluster size was slightly reduced (voxel-level p<0.001; cluster-level FWE-corrected p=0.054).

To provide a direct illustration of how the activation level in the right PPC varied according to the sequential distances in the two perspectives, we built another GLM with each sequential distance (i.e., 1–5) in each task as a separate condition. *Figure 4C* confirms that the activation level in the external-perspective task went up as the sequential distance increased, and the activation level in the internal-perspective task went down as the target event moved far away from the reference event where participants self-projected themselves.

These results suggest that the parietal cortex implements an egocentric representation of the event sequence, which varies with different perspectives.

## The right hippocampal head implemented consistent representations of sequential distance across external and internal perspectives

We did not find any regions in the hippocampus showing a significant interaction effect between *Task Type* and *Sequential Distance*. Instead, we found a region in the right hippocampal head in which the activation level positively correlated with the sequential distance when we combined external- and internal-perspective tasks (*Figure 4D*; voxel level p<0.001, cluster-level FWE-corrected p<0.05). The MNI coordinate of the center voxel was 25,–11, –15. *Figure 4E* further shows that we did not find that such a positive correlation with the sequential distance significantly differed in the external- and internal-perspective tasks (paired t(31) = –0.906, p=0.372).

We also illustrate the activation level in this hippocampal region of each sequential distance in external- and internal-perspective tasks, respectively (*Figure 4F*): the activation level in the right hippocampus tended to go up as the sequential distance increased regardless of the tasks.

These results suggest that the hippocampus implements an allocentric representation of the event sequence independent of the perspectives, contrary to the egocentric representation in the PPC.

## The right hippocampal body implemented the representation of event duration in the internal-perspective task

In the cortex, we did not detect any regions where the activation level showed a significant interaction effect between *Task Type* and *Duration* or a positive main effect of *Duration*. In the hippocampus, we also found no regions where the activation level showed a significant interaction effect between *Task Type* and *Duration*. Instead, we found a region in the right hippocampal body where the activation level showed a significantly positive correlation with *Duration* when combining two tasks (*Figure 5A*; voxel level p<0.001, cluster-level FWE-corrected p<0.05). The MNI coordinate of the center voxel is 39, –24, –12. However, *Figure 5B* shows that directly comparing the beta estimates in the two

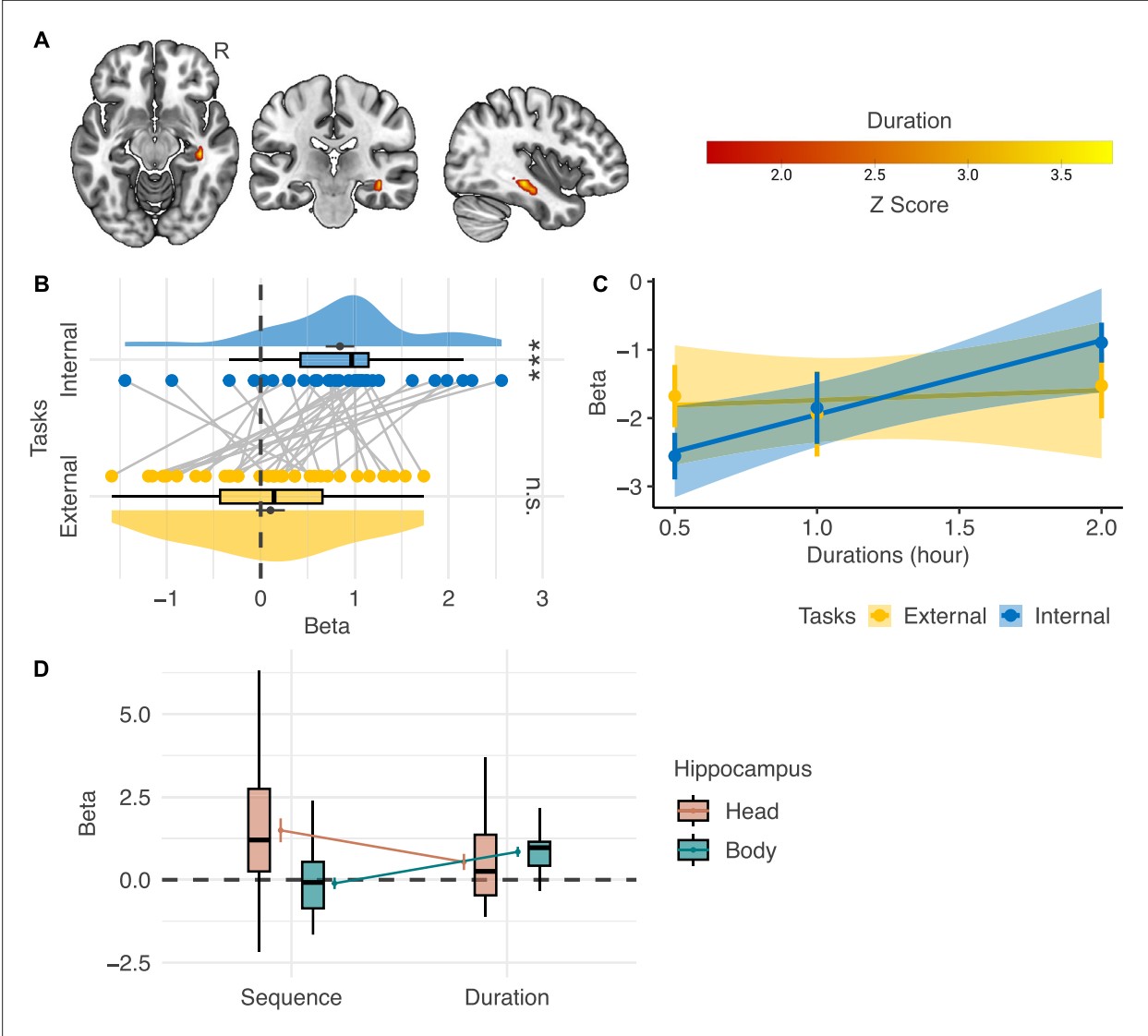

**Figure 5.** Neural correlates of event duration. (**A**) The right hippocampal body shows a significant main effect of *Duration* within the mask of the bilateral hippocampus (n=32; voxel-level p<0.001, cluster-level FWE-corrected p<0.05; voxel-level p<0.05 for illustration purposes). (**B**) However, regions of interest analysis shows that the correlation between the activation level in the right hippocampal body and *Duration* significantly differs in the internal- and the external-perspective task (n=32). (**C**) A further illustration of the relations between the activation level in the right hippocampal body and duration in the two tasks (n=32). The error bar indicates the standard error relative to the mean, and the shaded band around the linear regression line indicates 95% confidence interval. (**D**) Directly comparing the effects of *Sequential Distance* and *Duration* in the head and the body of the hippocampus shows a double dissociation pattern: the hippocampal head represented *Sequential Distance* but not *Duration*, while the hippocampal body represented *Duration* but not *Sequential Distance* (n=32). R=right hemisphere. ***p<0.001; n.s.: not significant.

The online version of this article includes the following source data for figure 5:

**Source data 1.** Thresholded Z-score maps of the neural correlates of duration and the extracted beta values for ROI analysis.

tasks reveals a significant result (paired t(31) = 3.07, p=0.004). The mean activation level in this area positively correlated with *Duration* only in the internal-perspective task (t(31) = 5.54, p<0.001), not in the external-perspective task (t(31) = 0.68, p=0.502). This later analysis is circular, as the ROI was defined as the voxels showing a significantly positive correlation with *Duration* combining two tasks in the first place. Nevertheless, it illustrated that this positive correlation in the voxel-level analysis was driven only by the internal-perspective task. Due to the stringent threshold of multiple comparisons across voxels, this interaction effect between *Task Type* and *Duration* was not found in the voxel-level analysis.

*Figure 5C* directly illustrates how the activation level in this hippocampal region varied according to duration: the activation level in the right hippocampal body went up as the duration increased only in the internal-perspective task but not in the external-perspective task.

The difference in duration representation between the two tasks remains open to interpretation. One possible explanation is that the hippocampus is preferentially involved in memory for durations embedded within event sequences (see review by *Lee et al., 2020*). In the internal-perspective task, participants indeed localized events within the event sequence itself. By contrast, the external-perspective task encouraged participants to compare the event sequence with external temporal landmarks, which may have attenuated the hippocampal representation of duration.

We also implemented a post hoc analysis to fully illustrate the *Sequential Distance* and *Duration* effects in the internal-perspective task within the regions of interest of the hippocampal *Head* and the *Body* (*Figure 5D*). These two regions were defined as the regions where activation level significantly positively correlates with *Sequential Distance* and *Duration,* respectively. No significant evidence shows that the hippocampal head also represented *Duration* (t(31) = 0.506, p=0.616), and no significant evidence shows that the hippocampal body also represented *Sequential Distance* (t(31) = –0.616, p=0.543). Further confirmatory analyses are needed to verify such a double dissociation pattern between the neural representation of sequence and duration in the head and the body of the hippocampus.

## Discussion

This study investigated the neural correlates of sequence and duration in task contexts where participants took internal or external perspectives on event series, thereby testing the complementary allocentric and egocentric reference frames in the temporal domain. We found that both the right PPC and the right hippocampal head represented the sequential distance between the reference and the target events. However, the representation in the right PPC varied with the perspective taken; its activity level correlated with sequential distance positively in the external perspective task and negatively in the internal perspective task. In contrast, the activation level in the right hippocampal head positively correlated with sequential distance regardless of the perspective taken. Moreover, we found that the activation level in the right hippocampal body positively correlated with the event duration in the internal-perspective task.

The negative correlation between the activation level in the right PPC and sequential distance has already been observed in a previous fMRI study by *Gauthier and van Wassenhove, 2016b*. In their study, the participants were instructed to mentally position themselves at a specific time point and judge whether a target event occurred before or after that time point. The authors identified a similar brain region (reported MNI coordinates of the peak voxel: 42, –70, 40), closely matching the activation observed in the present study (MNI coordinates of the peak voxel: 39, –70, 35). In both studies, activation in this region increased as the target event approached the self-positioned time point, which aligns with the evidence suggesting that the PPC implements egocentric representations. For example, neuropsychological studies have demonstrated that patients with lesions in the bilateral or unilateral right PPC have 'egocentric disorientation' (*Aguirre and D'Esposito, 1999*): they are unable to localize objects in relation with themselves (e.g., Case 2: *Levine et al., 1985*; Patient DW: *Stark, 1996*; Patient MU: *Wilson et al., 1997*; *Wilson et al., 2005*). Consistently, we found in a recent fMRI experiment that the distributed activity pattern in the bilateral PPC could encode egocentric but not allocentric directions of the target objects during memory retrieval (*Dutriaux et al., 2024*).

What is novel here is that the correlation between the activation level of the right PPC and sequential distance was reversed to a positive one when an external perspective was used. Since an external perspective is often associated with the allocentric reference frame (e.g., *O'keefe and Nadel, 1978*; *Arzy and Schacter, 2019*), our results seem to challenge the view that the parietal cortex implements egocentric representation. However, the conflict is more apparent than real. Perspective-taking, whether internal or external, by definition, must be egocentric, as mental images must be constructed from a specific viewpoint (see similar arguments by *Vogeley and Fink, 2003*; *Filimon, 2015*). Therefore, a brain region that implements egocentric representation should vary its activity with the perspectives taken, as shown in the right PPC. This finding is crucial; it suggests that the distance coding in the right PPC is not a 'perspective agnostic' magnitude effect, a possibility that previous fMRI studies could not definitively rule out.

Several previous studies have already questioned the nature of the PPC's function (e.g., see reviews by *Andersen et al., 1997*; *Cohen and Andersen, 2002*; *Wagner et al., 2005*; *Abrahamse et al., 2014*; *Ciaramelli et al., 2008*; *Cabeza et al., 2008*; *Hutchinson et al., 2009*; *Whitlock, 2017*; *Sestieri et al., 2017*; *Summerfield et al., 2020*; *Bottini and Doeller, 2020*). Here, we want to highlight three crucial findings. First, neuropsychological studies indicate that lesions in the bilateral PPC or lateral occipitoparietal cortex will lead to not only egocentric disorientation but also simultanagnosia, one of the key components of Bálint's syndrome (*Bálint, Dr., 1909*; *Rizzo and Vecera, 2002*; *Chechlacz and Humphreys, 2014*; e.g., WF: *Holmes and Horrax, 1919*; MVV: *Kase et al., 1977*; MU: *Wilson et al., 1997*; *Wilson et al., 2005*). Patients with simultanagnosia cannot perceive multiple entities as a whole and comprehend the overall meaning of a scene. Second, electrophysiological recordings in the lateral intraparietal area of the macaque cortex find the gain-field neurons, which can be used to transform the reference frames anchored to different body parts (e.g., *Andersen et al., 1985*; *Zipser and Andersen, 1988*; *Andersen et al., 1997*; *Cohen and Andersen, 2002*). Third, the fMRI studies in humans show that the particular PPC area found in our study (i.e., the hIP5 area in the lateral wall of the intraparietal sulcus and the dorsal PGp in the angular gyrus) is more engaged in episodic memory retrieval, in contrast to the medial wall of the intraparietal sulcus and dorsal part of the PPC more involved in perceptual attention (e.g., *Hutchinson et al., 2009*; *Sestieri et al., 2010*; *Sestieri et al., 2017*). Taking all these findings together, the PPC area identified in this study might contribute to memory retrieval in egocentric reference frames, which maintains relations among multiple memorized entities in the working memory from a perspective optimal for the current task context (*Abrahamse et al., 2014*). Such relational schema, constructed from stereotypical perspectives, is also represented in the PPC as a template to guide attention (*Summerfield et al., 2020*; *Bottini and Doeller, 2020*). It is thus not surprising that the two most common perspectives for time construals—mental time travel and mental time watching—involve the PPC.

In contrast to the PPC, we observed that the activation level in the head of the right hippocampus positively correlated with the sequential distance, regardless of the perspectives. Previous studies have already shown that the hippocampal activation level correlates with distance (e.g., *Ezzyat and Davachi, 2011*; *Howard et al., 2014*; *Garvert et al., 2017*; *Theves et al., 2019*; *Viganò et al., 2023*), and the distributed activity in the hippocampus can encode distance (e.g., *Deuker et al., 2016*; ; *Park et al., 2021*). Most studies reported hippocampal effects either bilaterally or predominantly in the right hemisphere, whereas only one study identified the effect in the left hippocampus instead (i.e., *Ezzyat and Davachi, 2011*). Our study is novel in showing that this distance coding in the hippocampus is independent of perspectives, indicating an allocentric representation of the event series. This finding aligns with the hypothesis in the spatial domain suggesting the hippocampus as an allocentric cognitive map (e.g., *O'keefe and Nadel, 1978*; *Byrne et al., 2007*; *Bicanski and Burgess, 2018*; *Bicanski and Burgess, 2020*; *Bottini and Doeller, 2020*). The positive correlation between the sequential distance and the hippocampal activation level might be mediated by adaptation (e.g., *Grill-Spector and Malach, 2001*): the longer the distance, the less the adaptation, and the greater the hippocampal activation. One way to interpret such perspective-agnostic representation in the hippocampus is to view it as the associations among memorized entities (e.g., *Muller et al., 1996*; *Eichenbaum, 2014*; *Eichenbaum, 2017*; *Buzsáki and Tingley, 2018*; *Quian Quiroga, 2019*). As a result, the allocentric representation in the hippocampus may not require a 'reference frame' because there is no fixed reference point. Each entity is represented among the relations to others, contrasting with the egocentric representation in the parietal cortex, where the reference point is clearly the self.

In this context, the distinction between the allocentric and the egocentric representations of an event series can also be understood in terms of memory storage and retrieval (also see *Byrne et al., 2007*). The hippocampus stores event series in a static and perspective-agnostic manner, while the PPC flexibly retrieves memory by constructing egocentric images tied to variable perspectives. Supporting this hypothesis is evidence that bilateral hippocampal lesions result in memory deficit for event series (e.g., *Dede et al., 2016*), whereas bilateral PPC lesions impair free recall but do not cause memory loss (e.g., *Berryhill et al., 2007*). Patients can still recall details of past events when prompted with cues or questions but struggle to access memories spontaneously.

Finally, the study indicates that the event durations were represented in the right hippocampal body, in line with previous studies suggesting that the hippocampus contributes to representing durations that are embedded within the event sequence (e.g., *Barnett et al., 2014*; *Thavabalasingam*

*et al., 2018*; *Lee et al., 2020*). Notably, the post hoc analysis reveals a double dissociation pattern: the hippocampal head represented sequential distances between events, whereas the hippocampal body represented the duration of individual events. Evidence from the spatial domain has suggested that the anterior hippocampus (or the ventral rodent hippocampus) implements global and gist-like representations (e.g., larger receptive fields), whereas the posterior hippocampus (or the dorsal rodent hippocampus) implements local and detailed ones (e.g., finer receptive fields) (e.g., *Jung et al., 1994*; *Kjelstrup et al., 2008*; *Collin et al., 2015*; see reviews by *Poppenk et al., 2013*; *Robin and Moscovitch, 2017*; see *Strange et al., 2014* for a different opinion). Recent evidence further shows that the organizational principle observed along the hippocampal long axis may also extend to the temporal domain (*Montagrin et al., 2024*). In that study, the anterior hippocampus showed greater activation for remote goals, whereas the posterior hippocampus was more strongly engaged for current goals, which are presumed to be represented in finer detail. Thus, the hippocampus likely represents the sequence of events hierarchically along its longitudinal axis: from the gist event sequence to the event sequence constituting a gist event, down to the unitary time bin sequence constituting an event as its duration. This view unifies the hippocampus's role in sequence representation and explains why it only represents durations embedded within event sequences. Future studies are required to confirm this hypothesis and investigate whether such hierarchical structures along the longitude axis of the hippocampus can also be used to represent the events at different time scales (e.g., from years to seconds). It would also be intriguing to examine whether the parietal cortex has a similar hierarchical structure or plays a role in zoom-in and zoom-out events at different scales to maintain an appropriate working memory load.

One limitation of the present study is that the external- and internal-perspective tasks differed not only in the type of perspective-taking they were intended to elicit, but also in their underlying decision-making processes. The external-perspective task explicitly required participants to compare two events with respect to external temporal landmarks and judge whether they occurred in the same or different parts of the day (i.e., a same/different judgment), whereas the internal-perspective task explicitly required participants to project themselves into a reference event and judge whether the target event occurred in the future or the past relative to that reference (i.e., a temporal-order judgment). This task design ensured that participants adopted two distinct perspectives on the event series, but at the expense of coherence in the cognitive operations required to make the two types of judgments. One alternative approach would be to more closely align the response demands of the two tasks by drawing on the A-series and B-series distinction (*McTaggart, 1908*): in the external-perspective task, participants could judge whether the target event occurred before or after the reference event (i.e., a before/after judgment), whereas in the internal-perspective task they could judge whether the target event occurred in the past or future relative to the reference event (i.e., a past/future judgment). Although such a design would improve coherence in the underlying decision-making processes (i.e., both are temporal-order judgments), it would reduce experimental control over the perspective-taking manipulation. For example, before/after judgments could still be made from an internal perspective. Future studies are therefore needed to determine whether findings obtained from these two task designs converge.

To conclude, this study reveals the neural correlates of sequence and duration in time construals. The hippocampal head represents the event sequence allocentrically, irrespective of the observer's perspectives, whereas the hippocampal body represents the event duration embedded in the event sequence. The PPC flexibly constructs the egocentric representation of the event series that varies according to the observer's perspectives, which could be internal (i.e., mental time travel) or external (i.e., mental time watching). Such allocentric and egocentric representations of event series can be interpreted in terms of memory storage and retrieval.

## Materials and methods
### Participants

Thirty-five native Italian speakers with no history of neurobiological or psychiatric disorders participated in the experiment. The sample size was chosen to align with the upper range of participant numbers reported in previous fMRI studies that successfully detected sequence or distance effects in the hippocampus (N=15–34; e.g., *Ezzyat and Davachi, 2011*; *Deuker et al., 2016*; *Garvert et al.,*

*2017*; *Howard et al., 2014*; *Theves et al., 2019*; *Park et al., 2021*; *Cristoforetti et al., 2022*). The ethical committee of the University of Trento approved the experimental protocol (Approval Number 2019-018), and all participants provided written informed consent and were paid for their time. Three participants were excluded. Two were due to poor behavioral performance during scanning; their accuracy was below 1.5 times the interquartile range below the lower quartile across participants. One was due to excessive head motion during scanning; this participant's mean frame displacement index (*Power et al., 2014*) was above 1.5 times the interquartile range above the upper quartile across participants. The remaining 32 participants entered the analysis (19 females, 13 males; age: 19–34; M=24.0; SD = 3.8; all participants except one were right-handed).

## Stimuli

We created a fictional religious ritual as the stimulus. Like most rituals, it follows a specific sequence, endures particular durations, and happens on predetermined parts of the day (*Figure 1A*). This created ritual comprised 15 events, falling into three parts of the day, that is, morning, afternoon, and evening. Each part of the day included five events and lasted 6 h in total: two events lasted for half an hour, one for 1 h, and the other two for 2 h. We instantiated each event with an event phrase. To minimize the potential confounding between the time information of the events and the semantic information of the phrases, we randomly assigned 15 phrases to the events twice, generating two versions for even and odd numbers of participants.

## Procedures

The day before fMRI scanning, participants learned the temporal information of the ritual: the sequence, the durations, and the parts of the day. The learning procedure included two phases. The first was the reading phase, and the second was the imagination phase. The two phases combined were performed twice.

In the reading phase, participants read a narrative describing the whole ritual on a computer screen twice (*Figure 1—source data 1*). The computer screen presented one sentence at a time, and each sentence provided information on one event. The presentation sequence was identical to the event sequence of the fictional ritual, and the sentences described the duration and the parts of the day of each event. Participants read the sentences at their own pace by pressing the spacebar to read the following sentence.

In the imagination phase, participants imagined themselves performing the whole ritual one event after another, guided by the prerecorded auditory instructions delivered through the headphones. Each event started with a voice telling the event to be imagined (e.g., 'light some candle'). A single beep was then played to indicate the start of the imagination, and a double beep sound indicated the end of the imagination. The imagining sequence was the same as the actual sequence, the imagining duration (i.e., the interval between the single and the double beep) was proportional to the actual duration (30 s/1 h), and the parts of the day were indicated in the instruction (e.g., 'the morning starts'). Participants were told to close their eyes to avoid distractions and imagined the whole ritual four times in each imagining phase (i.e., eight times in total).

Here, we let participants learn the event information through narrative reading or imagination. Compared to learning through actual experience, this approach prioritizes experimental control and efficiency. The timing of the events is compressed, akin to the process of retrospectively recalling our experiences, in which we mentally traverse events without requiring the actual time they originally took. However, future studies may be needed to investigate whether the encoding of events from first- and second-hand experience differs.

After learning, participants' knowledge of the ritual was assessed with three tests. In the event-sequence test, participants judged whether one event happened in the past or the future of another event. In the event-duration test, participants imagined the ritual in a self-paced manner and pressed the button when finishing the imagination of each event. In the parts-of-the-day test, participants judged whether two events happened in the same or different parts of the day. All participants' performance was greater than 80% in both the event-sequence and parts-of-the-day tests. In the event-duration test, Pearson's correlation between the self-paced imagining duration and the actual duration across 15 events was greater than 0.6 in all participants.

The scanning consisted of six runs, each with one external-perspective task block and one internal-perspective task block. The order of these two task blocks was interleaved across the six runs within each participant, and the order of these two task blocks in the first run was counterbalanced across participants. That means half the participants followed the order 'EI-IE-EI-IE-EI-IE', and half of the participants followed the order 'IE-EI-IE-EI-IE-EI' ('E' indicating the external-perspective task block and 'I' indicating the internal-perspective task block). Each task block started with a 5-s task prompt indicating the task of this block and a 3-s countdown presentation (i.e., the screen presenting '3', '2', '1'). The task block had 20 trials that were identical in the two task blocks within the same run but were presented in a randomized order.

In each trial, the phrases of two events were visually presented one after the other (*Figure 1C*). Participants first saw a 0.5 s fixation cross, the phrase of the reference event for 0.8 s, and a blank screen for 3.2 s (75% trials) or 7.2 s (25% trials). They then saw another 0.5 s fixation cross and the phrase of the target event. Each event phrase was presented in the center of a gray screen in two rows (black color, Calibri font): in the first row was the verb of the phrase (i.e., 'light'), and in the second row was the object of this verb 'some candle'. When presenting the target event phrase, we also presented two possible answers under the target event phrase; they were smaller in font size and colored differently depending on the task (i.e., red for the external-perspective task and blue for the internal-perspective task). In the external-perspective task block, the participants judged whether the target event happened in the same or different part of the day of the reference event, and the two option words were 'same' and 'different'. In the internal-perspective task block, the participants were instructed to project themselves into the reference event and judge whether the target event happened in the past or will happen in the future, and the two option words were 'past' and 'future'.

Participants were instructed to perform the two tasks once they saw the target event phrase by pressing two buttons with their right index and middle fingers, corresponding to the left and the right options on the screen, respectively. The RT was calculated as the duration between the onset of the target event phrase and the button press. A blank screen would replace the target event presentation once the response was given, and the total duration of the target event and the blank screen was maintained as 4 s. On 75% of occasions, the subsequent trial started immediately; on the other 25%, there would be another 4 s blank screen between trials.

We carefully selected two sets of 20 event pairs from the 210 possible combinations, assigning them to the odd and even runs of the fMRI experiment. Using a brute-force search, we identified 20 pairs in which sequential distance showed only weak correlations with positional information for both reference and target events (ranging from 1 to 15), as well as with behavioral responses (Same vs. Different and Future vs. Past, coded as 0 and 1), with all correlation coefficients below 0.2. At the same time, we balanced the proportion of correct responses across conditions: for the external-perspective task, Same/Different = 11/9 and 12/8; for the internal-perspective task, Future/Past = 12/8 and 8/12. Under these constraints, the sequential distances in both sets ranged from 1 to 5. To further mitigate spatial response biases, we pseudorandomized the left/right on-screen positions of the two response options within each task block, while ensuring an equal number of correct responses mapped to the left and right buttons (i.e., 10 per block).

## Behavior analysis

To investigate the factors affecting trial-by-trial RT of the correct trials, we built LMMs with *Participant* as the random effects grouping factor using the lme4 package (*Bates et al., 2015*). We fit the maximal model including the random intercept and all the random slopes consistent with the experimental design (as recommended by *Barr et al., 2013*). In the case of overfitting (singular fit), we removed the random slopes but kept the random intercept (*Matuschek et al., 2017*).

## MRI acquisition

MRI data were acquired using a MAGNETOM Prisma 3T MR scanner (Siemens) with a 64-channel head-neck coil at the Centre for Mind/Brain Sciences, University of Trento. Functional images were acquired using the simultaneous multislice echoplanar imaging sequence: the scanning plane was parallel to the long axis of the hippocampus, the phase encoding direction was from anterior to posterior, repetition time (TR)=1000 ms, echo time (TE)=28 ms, flip angle (FA)=59°, field of view (FOV)=200

mm × 200 mm, matrix size = 66 × 66, 65 axial slices, slices thickness (ST)=3 mm, no gap, voxel size = 3.03 × 3.03×3 mm, multiband factor = 5.

Field maps were acquired between each pair of functional runs to correct for geometric distortions of these two runs. Each set of field maps included three images, one phase-drift image between two slightly different echo times and two magnitude images for each of these two echo times: the scanning plane was parallel to the long axis of the hippocampus, the phase encoding direction was from anterior to posterior, TR = 1030 ms, shorter TE = 4.92 ms, longer TE = 7.38 ms, FA = 60°, FOV = 210 mm × 210 mm, matrix size = 70 × 70, 66 axial slices, ST = 3 mm, no gap, voxel size = 3 × 3 × 3 mm.

Three-dimensional T1-weighted images were acquired using the magnetization-prepared rapid gradient-echo sequence, sagittal plane, TR = 2530 ms, TE = 1.69 ms, inversion time = 1100 ms, FA = 7°, FOV = 256 mm × 256 mm, matrix size = 256 × 256, 176 continuous sagittal slices, ST = 1 mm, voxel size = 1 × 1 × 1 mm.

## MRI preprocessing

We preprocessed the brain images using SPM12 (https://www.fil.ion.ucl.ac.uk/spm/software/spm12/). The functional images were first realigned to the first image in the first run, generating six rigid head motion parameters for each time point and a mean functional image across all the runs. We then used the field maps to calculate the voxel displacement maps and coregistered them to this mean functional image. Since we acquired the field maps between each pair of functional runs, we applied each voxel displacement map to its two closest functional runs to correct their geometric distortions. The resulting functional images were next normalized to the MNI space with the acquired T1-weighted image using the unified segmentation method. In the final step, we spatially smoothed the normalized functional images, and the full width at half maximum of the 3D Gaussian smoothing kernel was 4 mm.

## First-level fMRI analysis

We performed the first-level analysis using SPM12 (https://www.fil.ion.ucl.ac.uk/spm/software/spm12/). GLMs were built to predict each participant's blood-oxygen-level-dependent (BOLD) signal. The GLMs in the primary analysis included six conditions: the task prompt and the countdown at the beginning of each block, the reference event and the target event in the external-perspective task, and the reference event and the target event in the internal-perspective task. The duration of the task prompt and the countdown was set as the duration of the actual presentation (i.e., 5 s and 3 s, respectively). The duration of the four event conditions was set as 0. The resulting boxcar and stick functions were convolved with a canonical hemodynamic response function (HRF). Head motion parameters and constant variables indicating each of the six runs were included as nuisance regressors. A high-pass filter with a cutoff of 128 s was used to remove the low-frequency noise and slow signal drifts.

We used the parametric modulation method to investigate the two core temporal variables: sequence (i.e., the number of events between the reference event and the target event) and duration (i.e., the duration of the target events in terms of hours). The z-scores of these parameters across the events in each run were set as parameters modulating the target events in both external- and internal-perspective tasks. The option for orthogonalizing modulations in the SPM was turned off (*Mumford et al., 2015*).

As a sanity check, we investigated whether the parametric modulation method in this study can successfully detect the visual cortex whose activation should be modulated by the number of syllables of the visually presented event phrase. To this end, we built a stick function with the sticks located on the onsets of both reference and target events and the 'height' of the stick as the z score of the number of syllables across the event phrases in each run. This stick function was convolved with a canonical HRF as an additional regressor involved in the GLM.

To detect the specific activations for external- and internal-perspective tasks, we directly contrasted the target event in the external-perspective task and the target event in the internal-perspective task. To investigate how the neural activation was modulated by each of the temporal information across different perspectives, we looked at its interaction effect with the task (i.e., contrast the effect between external- and internal-perspective tasks, i.e., contrast weights vector: 1, –1) and the main effect regardless of tasks (i.e., contrast weights vector: 1, 1).

To validate whether any significant effects were explained by the RT, we also built the same GLM incorporating trial-by-trial RT as a covariate. To do so, we created a stick function with the sticks

positioned at the onset of the target events in both external- and internal-perspective tasks, and the height of each stick was set to the z-score of the corresponding RT. These sticks were convolved with a canonical HRF to serve as a regressor in the GLM.

To directly illustrate how the activation level varied with the sequential distance, we built an independent GLM with each distance (i.e., from 1 to 5) in each task as a separate condition (i.e., 10 conditions in total). To directly illustrate how the activation level varied with duration, we built another GLM with each duration (i.e., 0.5, 1, and 2 h) in each task as a separate condition (i.e., six conditions in total). In both GLMs, we involved the task prompt and the countdown as conditions of no interest. The duration of the task prompt and the countdown was set as the duration of the actual presentation (i.e., 5 s and 3 s, respectively). The duration of all the other conditions was set as 0. The resulting boxcar and stick functions were convolved with the canonical HRF. Head motion parameters and constant variables indicating each of the six runs were included as nuisance regressors. A high-pass filter with a cutoff of 128 s was used to remove the low-frequency noise and slow signal drifts.

### Second-level fMRI analysis

We performed the group-level one-sample test on the first-level beta images from the univariate contrast and parametric modulation analyses. The statistical inference was performed using the permutation method with the PALM toolbox (https://fsl.fmrib.ox.ac.uk/fsl/fslwiki/PALM). 5000 sign flips were performed (*Winkler et al., 2014*), and a generalized Pareto distribution was fit to model the tail of the permutation distribution for the p values below 0.1 (*Knijnenburg et al., 2009*; *Winkler et al., 2016*). We controlled the family-wise error rate using a conventional cluster-forming threshold (i.e., voxel-wise p<0.001, cluster-level FWE-corrected p<0.05) (*Woo et al., 2014*). Given that the hippocampus is our primary region of interest, its elongated and thin shape limits the number of contiguous voxels, restricting the formation of large clusters, and functionally independent clusters within the hippocampus may naturally be small, we performed multiple comparison corrections separately for the cortex and the hippocampus. We used the Automated Anatomical Labelling Atlas 3 to define these masks (*Rolls et al., 2020*). The cortical mask was defined as all the cortical regions in both hemispheres combined, and the hippocampal mask was defined as the hippocampus in both hemispheres combined (i.e., the No. 41 and the No. 42 areas combined).

## Acknowledgements

This work was supported by the European Research Council (ERC-StG, NOAM 804422) and the Italian Ministry of University and Research (MUR-FARE, MODGET R18WJMSNZF) attributed to RB. We thank Mattia Silvestri for the help with data collection and Simone Viganò for the discussion.

## Additional information

### Competing interests

Roberto Bottini: Reviewing editor, *eLife*. The other authors declare that no competing interests exist.

### Funding

| Funder | Grant reference number | Author |
|---|---|---|
| European Research Council | 804422 | Roberto Bottini |
| Italian Ministry of University and Research | R18WJMSNZF | Roberto Bottini |

The funders had no role in study design, data collection and interpretation, or the decision to submit the work for publication. Open access funding provided by Max Planck Society.

### Author contributions

Yangwen Xu, Conceptualization, Data curation, Formal analysis, Investigation, Visualization, Methodology, Writing – original draft, Writing – review and editing; Nicola Sartorato, Conceptualization,

Data curation, Formal analysis, Investigation, Writing – review and editing; Léo Dutriaux, Conceptualization, Investigation, Visualization, Writing – review and editing; Roberto Bottini, Conceptualization, Resources, Supervision, Funding acquisition, Investigation, Writing – original draft, Project administration, Writing – review and editing

### Author ORCIDs
Yangwen Xu https://orcid.org/0000-0003-1482-4927
Nicola Sartorato https://orcid.org/0009-0002-9416-7954
Léo Dutriaux https://orcid.org/0000-0001-6304-8691
Roberto Bottini https://orcid.org/0000-0001-7941-7762

### Ethics
Human subjects: Informed consent and consent to publish were obtained. The ethical committee of the University of Trento approved the experimental protocol (Approval Identifier: 2019-018).

Reviewer #2 (Public review): https://doi.org/10.7554/eLife.107273.4.sa1
Author response https://doi.org/10.7554/eLife.107273.4.sa2

---

## Additional files

### Supplementary files
MDAR checklist

Supplementary file 1. Reaction time analysis including parts of the day.

### Data availability
Figure 2-source data 1, Figure 3-source data 1, Figure 4-source data 1, and Figure 5-source data 1 contain the data used to generate the corresponding figures.

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
