## [Editor Report · eLife Assessment]

This study presents a **valuable** finding on the neural representation of time from two distinct egocentric and allocentric reference frames. The evidence is **solid** and largely supports the hypothesis, with one caveat that the task differences could impact the observed effects. The work will be of interest to cognitive neuroscientists working on the perception and memory of time.

---

## [Referee Report · Reviewer #2 (Public review)]

Summary:

Xu et al. used fMRI to examine the neural correlates associated with retrieving temporal information from an external compared to internal perspective ('mental time watching' vs. 'mental time travel'). Participants first learned a fictional religious ritual composed of 15 sequential events of varying durations. They were then scanned while they either (1) judged whether a target event happened in the same part of the day as a reference event (external condition); or (2) imagined themselves carrying out the reference event and judged whether the target event occurred in the past or will occur in the future (internal condition). Behavioural data suggested that the perspective manipulation was successful: RT was positively correlated with sequential distance in the external perspective task, while a negative correlation was observed between RT and sequential distance for the internal perspective task. Neurally, the two tasks activated different regions, with the external task associated with greater activity in the supplementary motor area and supramarginal gyrus, and the internal condition with greater activity in default mode network regions. Of particular interest, only a cluster in the posterior parietal cortex demonstrated a significant interaction between perspective and sequential distance, with increased activity in this region for longer sequential distances in the external task but increased activity for shorter sequential distances in the internal task. Only a main effect of sequential distance was observed in the hippocampus head, with activity being positively correlated with sequential distance in both tasks. No regions exhibited a significant interaction between perspective and duration, although there was a main effect of duration in the hippocampus body with greater activity for longer durations, which appeared to be driven by the internal perspective condition. On the basis of these findings, the authors suggest that the hippocampus may represent event sequences allocentrically, whereas the posterior parietal cortex may process event sequences egocentrically.

Strengths:

The topic of egocentric vs. allocentric processing has been relatively under-investigated with respect to time, having traditionally been studied in the domain of space. As such, the current study is timely and has the potential to be important for our understanding of how time is represented in the brain in the service of memory. The study is well thought out and the behavioural paradigm is, in my opinion, a creative approach to tackling the authors' research question. A particular strength is the implementation of an imagination phase for the participants while learning the fictional religious ritual. This moves the paradigm beyond semantic/schema learning and is probably the best approach besides asking the participants to arduously enact and learn the different events with their exact timings in person. Importantly, the behavioural data point towards successful manipulation of internal vs. external perspective in participants, which is critical for the interpretation of the fMRI data. The use of syllable length as a sanity check for RT analyses as well as neuroimaging analyses is also much appreciated.

Suggestions:

The authors have satisfactorily addressed my last remaining suggestion.

---

## [Author Response]

The following is the authors’ response to the previous reviews

**Public Reviews:**

**Reviewer #1 (Public review):**
Summary:In this fMRI study, the authors wished to assess neural mechanisms supporting flexible temporal construals. For this, human participants learned a story consisting of fifteen events. During fMRI, events were shown to them, and participants were instructed to consider the event from "an internal" or from "an external" perspective. The authors found distinct patterns of brain activity in the posterior parietal cortex (PPC) and anterior hippocampus for the internal and the external viewpoint. Specifically, activation in the posterior parietal cortex positively correlated with distance during the external-perspective task, but negatively during the internal-perspective task. The anterior hippocampus positively correlated with distance in both perspectives. The authors conclude that allocentric sequences are stored in the hippocampus, whereas egocentric sequences are supported by the parietal cortex.

We thank the reviewer for the accurate summary of our study.

Strengths:The research topic is fascinating, and very few labs in the world are asking the question of how time is represented in the human brain. Working hypotheses have been recently formulated, and the work tackles them from the perspective of construals theory.

We appreciate the reviewer's positive and encouraging comments.

Weaknesses:Although the work uses two distinct psychological tasks, the authors do not elaborate on the cognitive operationalization the tasks entail, nor the implication of the task design for the observed neural activation.

We thank the reviewer for bringing this issue to our attention. In the revised manuscript, we have added a paragraph to the Discussion acknowledging this potential limitation of the study. Please see our response below.

**Reviewer #1 (Recommendations for the authors):**
Overall, I thank the authors for providing clear responses and much-needed detail on their original work, which enables a better understanding of their perspectives. I still have some detailed questions about the reported work, which I provide below. It could help clarify the work for a more general audience and its replicability by the community.

We thank the reviewer for their positive evaluation of our previous revisions.

Main general concern:I have one remaining core concern, which I distill as being a very different take on the usefulness of task design with neuroimaging. This concern follows from the authors' response to my original comment, which suggested possible confounds in fMRI data analysis and interpretation, as differences in task design and behavioral outcomes were not incorporated in the analytical approach.The authors confirmed that "there is a substantial difference between the two tasks" but argue that these differences are not relevant seing that "the primary goal of this study was not to directly compare these tasks to isolate a specific cognitive component " However, the authors do perform such contrasts in their analysis (e.g. p. 10: "We first directly contrasted the activity level between external- and internal-perspective tasks in the time window of...") and build inferences on brain activation from them (e.g., p. 10: "Compared with the internal-perspective task, the externalperspective task specifically activated the...").To clarify, my original concern was not about comparing neural activity in response to the two tasks but about the brain activity generated by two distinct tasks, which aim to reveal fundamentally distinct neural processes. The authors' response raises several concerns about the theoretical, methodological and empirical foundation of the work that are beyond the scope of a single empirical study and too long to detail here. Cognitive neuroscience relies on tasks to infer neural processes; this is the fertile and essential ground for using behavior in neuroscience to get to a mechanistic understanding of brain functions (e.g., Krakauer et al., 2017). In short, task design is fundamental because it shapes what neural processes are being investigated. Any inferences about brain activity recorded while a participant performs a task result from manipulated variables that should be under the control of the experimenter. Acknowledging that two tasks are distinct is acknowledging that different (neural) processes may govern their resolution. My initial remark was meant to highlight that, from basic signal detection theory, a same/different task and a temporal order task may not yield the same kind of basic biases and decision-making processes; these are far below and more basic than the posited sophisticated representations herein (construals, perspective taking).In short, the general approach is far coarser than the level of interpretational granularity being pushed forward in the paper would suggest.

We greatly appreciate the reviewer’s comments and agree that this is a very fair point. We acknowledge that the two tasks differ in their underlying decision-making processes. In the revised manuscript, we have added a paragraph at the end of the Discussion to explicitly acknowledge this limitation and to outline possible avenues for future research (Page 23).

“One limitation of the present study is that the external- and internal-perspective tasks differed not only in the type of perspective-taking they were intended to elicit, but also in their underlying decision-making processes. The external-perspective task explicitly required participants to compare two events with respect to external temporal landmarks and judge whether they occurred in the same or different parts of the day (i.e., a same/different judgment), whereas the internalperspective task explicitly required participants to project themselves into a reference event and judge whether the target event occurred in the future or the past relative to that reference (i.e., a temporal-order judgment). This task design ensured that participants adopted two distinct perspectives on the event series, but at the expense of coherence in the cognitive operations required to make the two types of judgments. One alternative approach would be to more closely align the response demands of the two tasks by drawing on McTaggart’s (1908) A-series and Bseries distinction: in the external-perspective task, participants could judge whether the target event occurred before or after the reference event (i.e., a before/after judgment), whereas in the internal-perspective task they could judge whether the target event occurred in the past or future relative to the reference event (i.e., a past/future judgment). Although such a design would improve coherence in the underlying decision-making processes (i.e., both are temporal-order judgments), it would reduce experimental control over the perspective-taking manipulation. For example, before/after judgments could still be made from an internal perspective. Future studies are therefore needed to determine whether findings obtained from these two task designs converge.”

Additional clarifications:Intro/theoryIn this revised MS, the authors provided some clarifications of their theoretical perspective in the introduction. From my standpoint, the motivation remains insufficiently precise for a scientific report. Some theoretical aspects, such as construals or perspective taking remain evasive in relation to ego and allocentric representations. A couple of paragraphs dedicated to explaining what the authors mean precisely when using these terms would greatly help to situate the validity of the working hypothesis. In the absence of clear definitions, it remains difficult to evaluate what is being tested. For instance, what do the authors mean by "time construal"? How is a time construal the same or not as a "temporal distance" or a "temporal sequence"? This would greatly help the readership.Additionally, some assertions are not clearly identified or fairly attributed. For instance, the assertion that EST provides a means to spatialize time is the authors' point of view or interpretation of this work, not an original proposition of the theory. Another example is McTaggart's metaphysics on time series (in the ontology of time in physics) "echoed" in linguistics; it has effectively been proposed and popularized by L. Boroditskty. The prospective and retrospective views of time should not be attributed to Tsao et al but to Hicks or Block in the 70's, who studied the psychology of time in humans.

We sincerely thank the reviewer for this criticism, which prompted us to clarify the relevant concepts in our manuscript. In the revised version, we made the following three main changes to the Introduction.

In the second paragraph of the Introduction (page 3), we clarify that event segmentation theory is independent of, but related to, the spatial construal of time hypothesis. We also clarify what we mean by time construals and explain that the two temporal components—duration and sequence—can be represented within such time construals, rather than constituting time construals themselves. These revisions were intended to prevent potential misunderstandings for the reader. In addition, we incorporated Boroditsky’s contributions relevant to this framework:

“One solution, which might be unique to humans, is to conceptualize time in terms of space (i.e., the spatial construal of time; e.g., Clark, 1973; Traugott, 1978; Lakoff & Johnson, 1980). Within this framework, time is usually first segmented into events—the basic temporal entities that observers conceive as having a beginning and an end (Zacks & Tversky, 2001). These temporal entities are then ordered in space, such that events occurring at different times can be maintained in working memory, allowing them to be flexibly accessed from different perspectives and easily referenced during communication (e.g., Casasanto & Boroditsky, 2008; Núñez & Cooperrider, 2013; Bender & Beller, 2014; Abrahamse et al., 2014; Figure 1A). The two core temporal components—duration and sequence—can be readily represented in such time construals.”

In the third paragraph of the Introduction (pages 3-4), we acknowledge the contributions of earlier behavioral studies on prospective and retrospective timing by citing the work suggested by the reviewer (Block & Zakay, 1997), which indicates that two distinct cognitive systems underlie timing processes. These behavioral findings converge with the conclusions of more recent neuroimaging studies:

“Unlike prospective timing tracking the continuous passage of time, durations in time construals are event-based (Sinha & Gärdenfors, 2014): the interval boundaries are constituted by events, and the event durations reflect their span (Figure 1A). Accumulating evidence suggests that distinct cognitive systems underlie these two types of duration (e.g., Block & Zakay, 1997). The motor and attentional system—particularly the supplementary motor area—has been associated with prospective timing (e.g., Protopapa et al., 2019; Nani et al., 2019; De Kock et al., 2021; Robbe, 2023), whereas the episodic memory system—particularly the hippocampus—is considered to support the representation of duration embedded within an event sequence (e.g., Barnett et al., 2014; Thavabalasingam et al., 2018; see also the comprehensive review by Lee et al., 2020).”

Block, R. A., & Zakay, D. (1997). Prospective and retrospective duration judgments: A meta-analytic review. Psychonomic Bulletin & Review, 4(2), 184-197.

In the fifth paragraph of the Introduction (page 5), we added a sentence to clarify the relationship between allocentric and egocentric reference frames and perspective taking:

“However, the neural mechanisms that enable the brain to generate distinct construals of an event sequence remain largely unknown. Valuable insights may be drawn from research in the spatial domain, which posits the existence of stable allocentric representations that are independent of viewpoint, from which variable egocentric representations corresponding to different perspectives can be generated.”

Methods:While more detail is provided in the Methods, some additional detail would be helpful to enable the replication of this work. For instance,- The table reports a sequence of phrases with assigned durations. Are the event phrases actual sentences given to participants? If so, how were participants made aware of the duration of the events, seeing that these sentence parts do not provide time information?

We apologize that we did not make this clear. The full text used during the reading phase of learning has already been provided in Figure 1—source data 1, which includes the information about event durations. In the revised manuscript, we now explicitly refer to this information in the Methods section (page 38): In the reading phase, participants read a narrative describing the whole ritual on a computer screen twice (Figure 1—source data 1).

- One of my original questions was about the narrative. In the Methods section, the authors state that participants read a text. Providing the full text would be helpful, also as a sanity check for sequentiality.

As clarified in the previous response, the texts are provided in Figure 1—source data 1, which illustrates the texts for both even- and odd-numbered participants.

- In the imagination phase, the authors introduce proportionality between imagination and experience (p. 37). What scale was used? What motivated it?

We thank the reviewer for bringing this issue to our attention. In this study, participants did not directly experience the events; instead, they learned the event information through narrative reading or imagination to ensure experimental control and efficiency. As clarified in the Methods section, the ratio between imagination duration and actual event duration was 30 seconds to 1 hour. In the revised manuscript, we have further explained our motivation for this design choice (page 39):

Here, we let participants learn the event information through narrative reading or imagination. Compared to learning through actual experience, this approach prioritizes experimental control and efficiency. The timing of the events is compressed, akin to the process of retrospectively recalling our experiences, in which we mentally traverse events without requiring the actual time they originally took. However, future studies may be needed to investigate whether the encoding of events from first- and second-hand experience differs.

Results:- p. 10: the interpretation of the data on chunking and boundary effects should be properly referenced to e.g. Davachi's published work.

We thank the reviewer for highlighting Davachi’s important work on event boundaries. We have appropriately cited these studies in the revised manuscript (page 10), as reflected in the following passage: This pattern can be interpreted as a categorical effect: sequential distances within the same part of the day were perceived as shorter (i.e., a chunking effect), whereas distances spanning different parts of the day were perceived as longer (i.e., a boundary effect). Similar boundary- or chunking-related effects on event cognition have been reported in previous studies (e.g., Ezzyat & Davachi, 2011; DuBrow & Davachi, 2013; Radvansky & Zacks, 2017).

Ezzyat, Y., & Davachi, L. (2011). What constitutes an episode in episodic memory?. Psychological Science, 22(2), 243-252.

DuBrow, S., & Davachi, L. (2013). The influence of context boundaries on memory for the sequential order of events. Journal of Experimental Psychology: General, 142(4), 1277.

Radvansky, G. A., & Zacks, J. M. (2017). Event boundaries in memory and cognition. Current Opinion in Behavioral Sciences, 17, 133-140.

**Reviewer #2 (Public review):**
Summary:Xu et al. used fMRI to examine the neural correlates associated with retrieving temporal information from an external compared to internal perspective ('mental time watching' vs. 'mental time travel'). Participants first learned a fictional religious ritual composed of 15 sequential events of varying durations. They were then scanned while they either (1) judged whether a target event happened in the same part of the day as a reference event (external condition); or (2) imagined themselves carrying out the reference event and judged whether the target event occurred in the past or will occur in the future (internal condition). Behavioural data suggested that the perspective manipulation was successful: RT was positively correlated with sequential distance in the external perspective task, while a negative correlation was observed between RT and sequential distance for the internal perspective task. Neurally, the two tasks activated different regions, with the external task associated with greater activity in the supplementary motor area and supramarginal gyrus, and the internal condition with greater activity in default mode network regions. Of particular interest, only a cluster in the posterior parietal cortex demonstrated a significant interaction between perspective and sequential distance, with increased activity in this region for longer sequential distances in the external task but increased activity for shorter sequential distances in the internal task. Only a main effect of sequential distance was observed in the hippocampus head, with activity being positively correlated with sequential distance in both tasks. No regions exhibited a significant interaction between perspective and duration, although there was a main effect of duration in the hippocampus body with greater activity for longer durations, which appeared to be driven by the internal perspective condition. On the basis of these findings, the authors suggest that the hippocampus may represent event sequences allocentrically, whereas the posterior parietal cortex may process event sequences egocentrically.

We sincerely appreciate the reviewers for providing an accurate, comprehensive, and objective summary of our study.

Strengths:The topic of egocentric vs. allocentric processing has been relatively under-investigated with respect to time, having traditionally been studied in the domain of space. As such, the current study is timely and has the potential to be important for our understanding of how time is represented in the brain in the service of memory. The study is well thought out and the behavioural paradigm is, in my opinion, a creative approach to tackling the authors' research question. A particular strength is the implementation of an imagination phase for the participants while learning the fictional religious ritual. This moves the paradigm beyond semantic/schema learning and is probably the best approach besides asking the participants to arduously enact and learn the different events with their exact timings in person. Importantly, the behavioural data point towards successful manipulation of internal vs. external perspective in participants, which is critical for the interpretation of the fMRI data. The use of syllable length as a sanity check for RT analyses as well as neuroimaging analyses is also much appreciated.

We thank the reviewer for the positive and encouraging comments.

Suggestions:The authors have done a commendable job addressing my previous comments. In particular, the additional analyses elucidating the potential contribution of boundary effects to the behavioural data, the impact of incorporating RT into the fMRI GLMs, and the differential contributions of RT and sequential distance to neural activity (i.e., in PPC) are valuable and strengthen the authors' interpretation of their findings.My one remaining suggestion pertains to the potential contribution of boundary effects. While the new analyses suggest that the RT findings are driven by sequential distance and duration independent of a boundary effect (i.e., Same vs. Different factor), I'm wondering whether the same applies to the neural findings? In other words, have the authors run a GLM in which the Same vs. Different factor is incorporated alongside distance and duration?

We thank the reviewer for their positive evaluation of our previous revisions and are pleased that the additional analyses adequately address the boundary effects in the behavioral data and the RT effects in the neural data.

With respect to boundary effects in the neural data, we followed the reviewer’s suggestion and constructed a more complex GLM that incorporated the Same/Different part of the day as an additional regressors modulating the target events. Importantly, the same PPC region continued to show an interaction effect between Task Type and Sequential Distance. We have added this important control analysis in our revised manuscript (Pages 13–14):

“To further assess whether the observed PPC reactivation can be attributed to boundary or chunking effects introduced by the Parts of the Day, as well as other behavioral outputs, we performed an additional control analysis. Using a more complex first-level model, we included two extra regressors modulating the target events in both internal- and external-perspective tasks, alongside Sequential Distance and Duration: (1) Same/Different parts of the day (coded as 1/−1) and (2) Future/Past (coded as 1/−1). Even with these additional controls, the same PPC region remained the strongest area across the entire brain, showing an interaction effect between Task Type and Sequential Distance, although the cluster size was slightly reduced (voxel-level p < 0.001; clusterlevel FWE-corrected p = 0.054).”